# Glioma and Peptidergic Systems: Oncogenic and Anticancer Peptides

**DOI:** 10.3390/ijms25147990

**Published:** 2024-07-22

**Authors:** Manuel Lisardo Sánchez, Arturo Mangas, Rafael Coveñas

**Affiliations:** 1Laboratory of Neuroanatomy of the Peptidergic Systems, Institute of Neurosciences of Castilla and León (INCYL), University of Salamanca, 37007 Salamanca, Spain; 2Grupo GIR USAL-BMD (Bases Moleculares del Desarrollo), University of Salamanca, 37007 Salamanca, Spain

**Keywords:** glioma, glioblastoma, peptides, peptide receptor, peptide receptor antagonist, peptidergic systems, oncogenic peptides, anticancer peptides

## Abstract

Glioma cells overexpress different peptide receptors that are useful for research, diagnosis, management, and treatment of the disease. Oncogenic peptides favor the proliferation, migration, and invasion of glioma cells, as well as angiogenesis, whereas anticancer peptides exert antiproliferative, antimigration, and anti-angiogenic effects against gliomas. Other peptides exert a dual effect on gliomas, that is, both proliferative and antiproliferative actions. Peptidergic systems are therapeutic targets, as peptide receptor antagonists/peptides or peptide receptor agonists can be administered to treat gliomas. Other anticancer strategies exerting beneficial effects against gliomas are discussed herein, and future research lines to be developed for gliomas are also suggested. Despite the large amount of data supporting the involvement of peptides in glioma progression, no anticancer drugs targeting peptidergic systems are currently available in clinical practice to treat gliomas.

## 1. Introduction

Peptidergic systems, via autocrine, paracrine, and endocrine mechanisms, are involved in the development of cancer. Promising anticancer strategies can be explored and developed by searching for compounds that specifically annihilate cancer cells by targeting these systems. Some peptides exert a proliferative effect on cancer cells, while others have an antiproliferative action. In some cases, the same peptide exerts antiproliferative or proliferative effects on different cancers [1]. Specific treatment against tumor cells is possible because, compared to healthy cells, cancer cells usually overexpress peptide receptors, which can be used as tumor biomarkers. Importantly, this overexpression has been associated with an increased risk of relapse, worse sensitivity to chemotherapeutic drugs, poor prognosis, larger tumor size, and higher aggressiveness [2,3]. The use of peptide receptor antagonists could be considered as a specific strategy to destroy tumor cells overexpressing peptide receptors. In fact, this strategy has shown beneficial effects against many types of cancers by promoting apoptosis in tumor cells and inhibiting their migration, invasion, and angiogenesis [1]. Additionally, peptide receptor antagonists in combination with radiotherapy or chemotherapy have promoted an antitumor synergic effect and significantly reduced the side effects induced by cytostatics [2,4].

The fifth edition of the World Health Organization (WHO) classification of tumors of the central nervous system, published in 2021 [5], classifies gliomas into ependymal, oligodendroglial, and astrocytic tumors with WHO grades 1–4, indicating different degrees of malignancy [6]. Glioblastomas (grade 4 gliomas) show a high invasive and proliferative capacity [7] and are the most common malignant brain and other central nervous system tumor in the USA, representing 50.9% of all malignant tumors [8]. At present, in addition to surgical resection, chemotherapy (e.g., temozolomide), and irradiation, new anti-glioma therapies have been tested, such as intranasal drug delivery, gene therapy, immunotherapy, tumor-treating fields, and microRNA [9,10,11,12,13,14]. Despite the current and new therapeutic procedures against gliomas, the survival time of patients after diagnosis is low, necessitating the urgent development of new therapeutic strategies to treat the disease. One such new anti-glioma therapeutic strategy is based on the involvement of peptidergic systems in glioma development, as many in vitro and in vivo experiments have demonstrated that some peptides exert an oncogenic effect on glioma cells while others promote anticancer action (see Section 2). Thus, this review focuses mainly on the involvement of peptidergic systems in glioma development, suggests anti-glioma therapeutic strategies targeting these systems, and highlights future research directions in gliomas. Unfortunately, no clinical trial has yet been performed to confirm anticancer strategies mediated by peptidergic systems against gliomas. However, as indicated below, many experiments have demonstrated the involvement of peptidergic systems in glioma progression, supporting the development of potential and promising anti-glioma therapeutic approaches in the future.

## 2. Glioma and Peptidergic Systems

In this section, we review the plethora of oncogenic and anticancer peptides that regulate glioma development. The full knowledge of the roles played by these peptides in glioma progression will serve to develop potential and promising anti-glioma therapeutic strategies.

### 2.1. Oncogenic Peptides

#### 2.1.1. Adrenomedullin

Adrenomedullin (AMD) is upregulated in several cancers (e.g., lung, prostate, colon, breast, and brain) [15]. AMD promotes glioblastoma cell growth, and when AMD expression is knocked down in glioma cells, the inhibitory action exerted by temozolomide and apoptotic mechanisms increases in these cells [16]. AMD is overexpressed in temozolomide-resistant glioma samples, and AMD knockdown augments the effects mediated by temozolomide on the Akt (protein kinase B), extracellular signal-regulated kinases (ERKs) 1/2, and Bax/Bcl-2 signaling pathways [16]. MicroRNA (miR)-1297 blocks AMD expression, sensitizes glioma cells to temozolomide treatment, and exerts similar actions in glioma cells treated with temozolomide as those reported in AMD knockdown glioma cells [16]. Importantly, AMD expression is positively associated with the grade of malignancy observed in gliomas, with the highest expression reported in glioblastoma [7]. Through activation of the ERK1/2 signaling pathway, AMD favors the formation of filopodia by tumor cells, augmenting their invasive capacity and increasing the proliferation of glioblastoma cells [7]. Moreover, the administration of monoclonal antibodies against AMD blocks tumor growth and enhances the antitumor effect exerted by temozolomide [7]. AMD also favors angiogenesis by acting on endothelial cells [17]. Thus, AMD, via autocrine and paracrine mechanisms, plays an important role in glioma progression. Tumor progression and angiogenesis induced by AMD are mediated by calcitonin receptor-like receptor/receptor activity-modifying proteins 2 and 3 [17]. 

Hypoxia occurs in the tumor microenvironment, and under hypoxic conditions, hypoxia-inducible factor 1 (HIF 1) regulates cellular responses [18]. Interleukin-1β, also located in the tumor microenvironment, controls the progression of glioblastoma, downregulates the expression of the HIF 1 target gene AMD, and blocks HIF 1 activity [18]. AMD counteracts the apoptotic mechanisms promoted by hypoxia in glioblastoma cells, and in the presence of interleukin-1β, the number of apoptotic tumor cells induced by hypoxic conditions increases [18]. This means that interleukin-1β promotes apoptosis in glioblastoma cells that synthesize AMD. The cytokine oncostatin, after inducing STAT-3 (signal transducer and activator of transcription-3) phosphorylation, promotes the expression of AMD in astroglioma cells, and this peptide also increases their migration [15]. This means that STAT-3 activation, which occurs in brain tumors, controls ADM expression and tumor cell invasion.

#### 2.1.2. Angiotensin

Glioblastoma cell lines (e.g., T98G and U-87MG) overexpress both angiotensin II peptide and angiotensin II type I receptors. The peptide, via angiotensin II type 2 receptors, induces the proliferation of tumor cells [19,20]. Glioma cell proliferation and invasion are also related to increased expression of angiotensin II type 1 receptors, and a decrease in the expression of these receptors, along with reduced proliferation and invasion of tumor cells, has been reported in glioblastoma cells overexpressing miR-155 [21]. miR-155 not only controls the expression of angiotensin II type 1 receptors but also attenuates angiogenesis and the angiotensin II type 1 receptor/CXCR4/NF-κB signaling pathway, promoting antitumor effects against glioblastoma cells [21]. This indicates that these inhibitors and miR-155 are promising therapeutic tools for treating glioblastoma cells expressing angiotensin II type 1 receptors. Telmisartan, an angiotensin II receptor blocker that also crosses the blood–brain barrier, inhibits the proliferation and migration/invasion of glioblastoma cells by inducing G0/G1 phase arrest/apoptosis and blocking tumor growth in vivo [22]. Angiotensin II augments the in vivo and in vitro PD-L1 (programmed death-ligand 1) expression in glioblastoma cells, which is counteracted by losartan [23]. This compound antagonizes the angiogenic and immunosuppressive mechanisms mediated by angiotensin II.

Gliomas express peptides of the renin–angiotensin system, which plays a crucial role in the glioblastoma microenvironment [24,25]. In this sense, the repurposing of drugs controlling this system has been suggested to treat glioblastoma [25]. Doxazosin, an antihypertensive drug, promotes the death of glioblastoma cells by blocking cell proliferation and inducing apoptosis [26]. The beneficial effects exerted by angiotensin system inhibitors and other antihypertensive compounds on the prognosis of recurrent glioblastoma patients treated with the antivascular endothelial growth factor bevacizumab have also been reported [27]. A phase I clinical trial using repurposed renin–angiotensin system modulators to treat glioblastoma showed promising results, although the number of patients treated was low (*n* = 17) [28]. In this trial, the overall median survival was higher than the current overall median survival, and all patients showed a preserved quality of life.

#### 2.1.3. Bombesin/Gastrin-Releasing Peptide/Neuromedin B

Gastrin-releasing peptide (GRP) and neuromedin B are bombesin-like peptides reported in mammals. Bombesin receptors are overexpressed in tumor cells and are involved in the proliferation of glioma cells [29,30,31]. Bombesin/GRP induces the proliferation of U-373MG glioblastoma cells and activates the mitogen-activated protein pathway [30]. Rodents with gliomas treated with the antitumor drug camptothecin show longer survival when the drug is delivered via bombesin/poly (ethylene glycol)/polycaprolactone-Tat mixed micelles than when camptothecin is delivered via poly (ethylene glycol)/polycaprolactone-Tat mixed micelles [32]. This highlights the importance of the chemical composition of micelles and the important role played by the bombesin/bombesin receptor system in glioma treatment. This system is also important for neurosurgeons to specifically identify tumor boundaries after using the specific fluorescent IRDye800-bombesin, which targets GRP receptors [33].

GRP receptors have been detected in human glioma samples [34]. These receptors control the proliferation of C6 glioma cells via a phosphatidylinositol 3-kinase (PI3K)-dependent mechanism [35]. GRP receptors mediate the proliferation of glioblastoma cells, and GRP receptor antagonists promote antitumor effects in in vitro and in vivo experimental models of glioblastoma [36]. GRP receptor knockdown induces senescence in glioblastoma cells, increases p16, p21, and p53 expression, decreases p38 expression, and activates epidermal growth factor receptors [36]. GRP has also been used for the delivery of irinotecan, a chemotherapeutic drug, to treat glioblastoma in in vitro and in vivo studies, showing an anticancer effect against U-87MG glioblastoma cells [37]. Finally, neuromedin B increases the expression of the c-fos gene, favors the release of arachidonic acid, and promotes growth in C6 glioma cells [38].

#### 2.1.4. Bradykinin

Bradykinin mediates the proliferation of glioblastoma cells via the activation of the ERK1/2/PIK3/Akt signaling pathway [39]. The bradykinin 1 receptor mediates the migration and invasion of glioblastoma cells and the synthesis of cytokines/chemokines in these cells [40]. Bradykinin stimulates the migration of glioma cells by upregulating the expression of cyclo-oxygenase 2 and activating the bradykinin 1 receptor and the PI3K/Akt, AP1, and c-Jun signaling pathways [41]. Bradykinin, via the bradykinin 1 receptor, promotes the expression of interleukin-8, leading to translocation into the nucleus of phosphorylated STAT3 and acetylated SP-1, favoring the migration of glioblastoma cells [42]. A previous study demonstrated that U-87MG glioblastoma cells treated with bradykinin increase the expression of bradykinin 1 receptors [43]. In the same study, bradykinin regulated aquaporin 4 gene expression and the migration and invasion of glioblastoma cells through the bradykinin 1 receptor, as well as the activation of the MEK1-ERK1/2-NF-κB signaling pathway [43]. Moreover, this receptor mediates tumor growth, upregulates PD-L1 expression in both macrophages and glioblastoma cells, and reduces survival rates in experimental animal models [40].

Bradykinin directs the invasion of glioma cells toward blood vessels via the bradykinin 2 receptor [44]. This is an important mechanism, as glioma cells are then exposed to cytokines, chemokines, growth factors, nutrients, and oxygen in the bloodstream, favoring the spread of glioma cells. Bradykinin augments the permeability of the blood–tumor barrier and regulates angiogenesis [45,46], and blood vessel permeability is regulated through the bradykinin 1 receptor [47]. Increased expression of the bradykinin 1 receptor has been reported in the glioblastoma microenvironment [46]. Crosstalk between mesenchymal stem cells (immunomodulatory and mobile tumor-tropic cells isolated from adipose tissue, the oral cavity, and other tissues) and glioblastoma cells has been reported in this microenvironment, and bradykinin and the bradykinin 1 receptor are involved in the interactions between these cells (cell–cell and cell fusion) [48]. The bradykinin 1 receptor favors the migration/invasion of tumor cells and mediates the fusion of glioblastoma and mesenchymal stem cells [48]. Due to the tumor tropism and endogenous origin of the latter cells, they are promising antiglioblastoma agents.

#### 2.1.5. Cholecystokinin

Glioblastoma cells (U-87MG) express both cholecystokinin (CCK) and CCK B receptors. The peptide promotes cell growth through autocrine mechanisms after binding to these receptors [49]. CCK antagonists block tumor growth by inducing apoptosis in tumor cells [49], augmenting the activity of caspase 3, and reducing proteasome activity.

#### 2.1.6. Endothelin

Endothelin 1 and endothelin A (ETA) receptors are overexpressed in tumors [50]. These receptors are overexpressed in glioblastoma compared to normal brain tissues, and administration of the ETA receptor antagonist BQ123 ameliorates tumor perfusion, benefiting nanomedicine delivery for treating glioblastoma [50]. Astrocytic tumors show very low expression of ETA receptors, whereas their expression is high in the tumor microvessels. Importantly, ETA expression increases gradually from grade 2 to grade 4 in tumor cells and microvessels [51]. ETB receptors mediate the proliferation of Hs683 human oligodendroglioma cells through an extracellular signal-regulated kinase mechanism. Treatment with a selective ETB receptor antagonist (BQ788) blocks oligodendroglioma cell proliferation, thereby enhancing survival [52].

#### 2.1.7. Neurotensin

The neurotensin 1 receptor is involved in glioma development. Neurotensin, acting via this receptor, promotes the proliferation and invasion of glioma cells [53,54]. Jun is positively regulated by the neurotensin 1 receptor, which mediates miR-494 upregulation and SOCS6 downregulation. Knockdown of the neurotensin 1 receptor in glioblastoma cells blocks tumor growth and invasion; however, miR-494 overexpression restores both effects [54]. Glioblastoma growth is inhibited in in vitro and in vivo experiments when the Wnt/β-catenin pathway or the neurotensin/neurotensin 1 receptor system is pharmacologically blocked [55]. The neurotensin 1 receptor antagonist SR48692 favors apoptosis in these cells, which is associated with the release of cytochrome C by mitochondria; however, neurotensin 1 receptor knockdown promotes slight apoptosis [56]. Moreover, the blockade of this receptor sensitizes glioblastoma cells to doxorubicin or antinomycin D and downregulates Bcl-2 and Bcl-w expression. These data suggest that the neurotensin 1 receptor is involved in protecting glioblastoma cells from intrinsic apoptosis.

#### 2.1.8. Substance P

The neurokinin 1 receptor (NK-1R) is overexpressed in astrocytoma (WHO grade 4), and its expression is essential for the viability of glioma cells (U-87MG and GAMG). This receptor shows a high affinity for the undecapeptide substance P [57]. The effects exerted by substance P labeled with the alpha emitter ^225^AC (^225^AC-DOTA-substance P) have been studied in glioblastoma cell lines (U-138MG, U-87MG, and T98G) and glioblastoma stem cells [58]. A significant reduction in tumor cell viability has been found after treatment with ^225^AC-DOTA-substance P due to the induction of apoptosis, as cells were arrested in the G2/M phase, and similar results were reported in glioblastoma stem cells [58]. A decrease in the number of glioma cells due to apoptotic and necrotic mechanisms has been observed after silencing the tachykinin 1 receptor. Moreover, the full-length NK-1R isoform is mainly observed in the nuclei of glioma cells, whereas the truncated NK-1R is mainly found in the cytoplasm of these cells [59]. NK-1R is expressed in glioma cells, and samples and substance P, via this receptor, favor the proliferation, migration, and invasion of glioma cells, as well as angiogenesis [60,61]. Meanwhile, NK-1R antagonists (e.g., aprepitant, L-732,138, and L-733,060) block angiogenesis and the proliferation/migration of glioma cells by promoting apoptotic mechanisms in tumor cells [60,61]. Accordingly, NK-1R antagonists (e.g., aprepitant) have been suggested to treat tumors, including gliomas [62], and hence, the substance P/NK-1R system is a promising antitumor target to treat gliomas. The antitumor effect of temozolomide, the antiviral drug ritonavir, and the brain-penetrant NK-1R antagonist aprepitant (an anti-emetic drug) is higher against GAMG glioma cells when co-administered than when separately administered or when the administration of temozolomide + aprepitant, ritonavir, or ritonavir + aprepitant is performed; thus, a significant antitumor synergic effect occurs when antiviral and anti-emetic drugs are co-administered with temozolomide [63]. Moreover, the substance P/NK-1R system is involved in molecular pathways associated with resistance, and it has been suggested that NK-1R antagonists could be useful in overcoming cancer resistance, although this must be confirmed in future studies [64]. The substance P/NK-1R system also regulates oxidative stress mechanisms in glioblastoma cells [43,65]. Substance P increases the levels of reactive oxygen species and malondialdehyde, decreases the level of thiol in the latter cells, and significantly reduces the total antioxidant cell capacity [43]. The selective NK-1R antagonist aprepitant decreases reactive oxygen species and malondialdehyde levels in glioblastoma cells, whereas an increase in antioxidant components of the redox system has been observed [43]. Moreover, aprepitant increases the expression of superoxide dismutase/catalase in glioblastoma cells; thus, aprepitant promotes antioxidant actions mediated by both enzymes [66]. 

Yokukansan, a traditional Japanese herbal medicine, blocks the synthesis of interleukins 6 and 8 induced by substance P in glioblastoma U-373MG cells, exerting anti-inflammatory effects [67]. Yokukansan downregulates COX-2 expression and suppresses ERK1/2 and p38 MAPK (mitogen-activated protein kinase) phosphorylation and NF-κB nuclear translocation, induced by substance P, in U-373MG glioblastoma cells [67]. Gabapentin and pregabalin, lipophilic amino acid derivatives of gamma-aminobutyric acid, block the activation of NF-κB induced by substance P in glioma cells [68].

### 2.2. Anticancer Peptides

#### 2.2.1. Angiotensin (1-7)

The angiotensin (1-7) fragment blocks glioblastoma growth and counteracts edema formation and blood–brain barrier damage, which is disrupted in gliomas [69]. In fact, this angiotensin fragment restores the expression of tight junction proteins (e.g., ZO-1 and claudin 5) by regulating the activation of the JNK signaling pathway. SP600125, a blocker of this pathway, augments the expression of tight junction proteins, reduces blood–brain barrier disruption, and increases the anti-glioma activity of angiotensin (1-7) [69].

#### 2.2.2. Carnosine

PHT1/2 and PEPT2 transporters mediate the uptake of carnosine into glioblastoma cells [70]. Carnosine blocks glioblastoma cell growth; this blockade is independent of the PI3K/Akt/mTOR (mammalian target of rapamycin) signaling pathway [71,72]. This dipeptide inhibits the proliferation, migration, and invasion of glioma cell lines (U-87MG and U-251MG) under hypoxic or normoxic conditions [73]. Importantly, carnosine prevents the formation of colonies of glioma cells and specifically destroys tumor cells in co-cultures with fibroblasts; the latter cells are healthy and survive [74]. Carnosine augments the antitumor efficacy of temozolomide and ionizing irradiation against glioblastoma cells, and the dipeptide protects non-tumor cells from irradiation [75]. This means that carnosine counteracts the side effects promoted by ionizing irradiation. Carnosine also blocks the anaerobic glycolytic metabolism in glioma cells, which is essential for the survival of tumor cells [76].

#### 2.2.3. Corticotropin-Releasing Factor/Urocortin

Corticotropin-releasing factor (CRF) and its receptors are expressed in many tumor types, including gliomas [77]. In gliomas, CRF mRNA levels are downregulated [78]. CRF inhibits the proliferation of glioma cells and favors long non-coding RNA-p21 expression, which leads to the suppression of both the proliferation and invasion of glioma cells [78]. The latter effect is blocked by miR-34c targeted by 3′-UTR, and miR-34c reduces the expression of CRF 1 receptors. CRF blocks glioma development by upregulating the long non-coding RNA-p21 [78]. It has been shown that the therapeutic efficacy of CRF depends on the CRF receptor type expressed by tumor cells [79].

Urocortin belongs to the CRF family of peptides. The mRNA expression of urocortin and CRF receptors has been detected in glioma cells [77]. CRF receptors have been observed in A172 glioblastoma cells but not in U-138MG glioblastoma cells, whereas urocortin I has been found in both glioblastoma cell lines [80]. Moreover, urocortin I is released from glioblastoma cells by exocytosis [80].

#### 2.2.4. Endothelin

Endothelin 1 and endothelin A (ETA) receptors are overexpressed in tumors [50]. These receptors are overexpressed in glioblastoma compared to normal brain tissues [50]. Astrocytic tumors show very low expression of ETA receptors, whereas this expression is high in tumor microvessels. Importantly, ETA expression increases gradually from grade 2 to grade 4 in tumor cells and microvessels [51]. ETB receptors mediate the proliferation of Hs683 human oligodendroglioma cells through an extracellular signal-regulated kinase mechanism [52].

High endothelin-converting enzyme 1 expression, involved in the activation of endothelin 1, is associated with glioblastoma development [81]. This enzyme increases tumor cell migration and invasion, and it has been suggested that endothelin-converting enzyme 1 could be a novel marker for poor prognosis as well as a new anti-glioma therapeutic target [81].

#### 2.2.5. Enkephalin

The opioid morphine favors human glioma T98G cell proliferation, whereas the opioid peptide analog biphalin decreases this proliferation [82]. Thus, it seems that biphalin can be used to treat pain and cancer at the same time. The hyperactivation of NFAT1, mediated by the pentapeptide methionine-enkephalin, promotes in vitro and in vivo apoptosis in rat C6 glioma cells [83]. The pentapeptide increases caspase 3, 8, and 9 activities and Bax, Fas, and FasL expression, decreases Bcl-2 expression, augments cytoplasmic Ca^++^ influx, and increases NFAT1 levels in the nucleus of C6 glioma cells [83]. NFAT1 can control the transcription of downstream genes (e.g., *FasL*), promoting apoptotic mechanisms. Methionine-enkephalin increases the cytotoxic activity of microglia against glioblastoma cells, increasing its antitumor efficacy [84]. Moreover, the peptide increases microglia phagocytosis capacity and also augments the inducible nitric oxide synthase and nitrite expression in these cells [84]. Thus, methionine-enkephalin regulates microglia functioning.

#### 2.2.6. Gonadotropin-Releasing Hormone

Glioblastoma cells express gonadotropin-releasing hormone (Gn-RH) receptors [85]. These receptors have been detected in glioblastoma cell lines (U-373MG and U-87MG) and in samples of patients suffering from the disease [86]. Gn-RH agonist zoladex (goserelin acetate), the first peptide-based antitumor drug approved by the Food and Drug Administration (FDA), decreases U-373MG/U-87MG/LN229 glioblastoma cell proliferation and increases kininogen 1 expression [85]. This means that a link between Gn-RH and the epidermal growth factor receptor signaling pathway through kininogen 1 is possible. The FDA has approved the use of Gn-RH peptide receptor agonists (e.g., zoladex, lupron (leuprolide), and triptorelin (trelstar)) and antagonists (e.g., plenaxis (abarelix) and firmagon (degarelix)) for the treatment of prostate and breast cancers, endometriosis, and uterine leiomyomata [87].

#### 2.2.7. Luteinizing Hormone-Releasing Hormone

U-87MG cells express luteinizing hormone-releasing hormone (LH-RH) receptors [88]. AN-152 (AEZS-108), an LH-RH analog, respectively decreased tumor growth and promoted apoptosis in glioblastoma U-87MG cells in in vivo and in vitro experiments [88].

#### 2.2.8. Pituitary Adenylate Cyclase-Activating Polypeptide

Pituitary adenylate cyclase-activating polypeptide (PACAP) binds to the selective PAC1 receptor as well as to vasoactive intestinal peptide (VIP) receptors (VPAC1 and VPAC2) [89,90]. The VPAC1 receptor is a useful target for positron emission tomography imaging of glioblastoma using the VPAC1-specific peptide [^64^Cu]TP3805. In addition, this procedure is a promising approach for glioblastoma research [91]. The expression of PAC1 receptors is upregulated in gliomas, particularly in oligodendrogliomas [89]. PACAP exerts an antiproliferative effect on glioblastoma cells [92], and PACAP-27 and PACAP-38, probably through VPC2 receptors, inhibit the proliferation of T98G glioblastoma cells [93].

PACAP agonists decrease the invasion of C6/U-87MG glioma cells [94]. This invasion is suppressed when PI3K/Akt and Sonic Hedgehog inhibitors are administered. PACAP regulates angiogenic pathways by blocking the formation of blood vessels and decreasing the release of vascular endothelial growth factor [95]. In addition, PACAP reduces the expression of mesenchymal markers (e.g., CD44, matrix metalloproteinases 2 and 9, and vimentin), affecting invasiveness [95]. PACAP-38 upregulates the expression of brain-derived neurotrophic factor in T98G glioblastoma cells [96]. High expression of PACAP receptors is correlated with the proliferative and malignant potential of astrocytoma cells [97].

#### 2.2.9. Somatostatin

Most gliomas overexpress the somatostatin 2 receptor; high expression of this receptor has been detected in oligodendrogliomas and astrocytomas, compared to a low level in glioblastomas [98,99]. Expression of the somatostatin 4 receptor has also been reported in glioblastomas [100]. Somatostatin suppresses the migration of glioma cells (U-87MG and T98G) without affecting Rac/PI3K activity, which are essential targets for motility regulation [101]. Somatostatin also inhibits the synthesis and release of angiogenic vascular endothelial growth factor from glioma cells [102].

Astrocytic tumors show very low expression of somatostatin 2, 3, and 5 receptors, whereas this expression is high in tumor capillaries. The truncated splicing variant of the somatostatin 5 receptor is overexpressed in glioblastoma cells, and it is related to malignancy [103]. This overexpression is associated with increased proliferation and migration of glioblastoma cells (U-87MG and U-118MG), recurrence, poor overall survival, alterations in the signaling pathways involved in tumor progression and aggressiveness (e.g., JAK-STAT, NF-κB, Akt), and somatostatin analog resistance, since the silencing of this somatostatin receptor variant sensitizes glioblastoma cells to the antitumor action exerted by the somatostatin analog pasireotide [103].

Antisomatostatin 2 receptor peptide-based targeted delivery of encapsulated 3,3′-diindolylmethane (antitumor compound) nanoparticles prevents the progression of gliomas; this procedure promotes apoptosis and abrogates the activation of the epidermal growth factor receptor pathway in glioma cells [104]. Endothelial cells also express the somatostatin 2 receptor [105]. Paclitaxel-loaded solid lipid nanoparticles modified with Tyr-3-octreotide (a ligand for the somatostatin 2 receptor) enhance antitumor and anti-angiogenic effects against gliomas, targeting tumor cells and neovasculature [105]. Many tumors overexpress growth hormones and growth hormone receptors; this system plays an important role in tumor development, which can be inhibited by somatostatin [106].

Somatostatin peptide analogs (e.g., depreotide, netspot, lutathera, detectnet, and somatuline) have been approved by the FDA for diagnostic/theragnostic purposes in tumors expressing somatostatin receptors (e.g., neuroendocrine and lung) [87].

#### 2.2.10. Tat-NTS/Tat-Cx43266-283

The Tat-NTS peptide inhibits the proliferation and migration/invasion of glioblastoma cells by blocking annexin-A1 nuclear translocation [107]. This peptide promotes cell cycle arrest, suppresses the NF-κB signaling pathway, and downregulates matrix metalloproteinase 2 and 9 expression [107]. The connexin 43 peptide Tat-Cx43266-283 also exerts antitumor action in preclinical, experimental glioblastoma models [108].

### 2.3. Oncogenic and Anticancer Peptides

#### 2.3.1. Growth Hormone-Releasing Hormone

Growth hormone-releasing hormone (GH-RH) acts as a growth factor in tumors [109]; hence, GH-RH antagonists have been used to treat tumors [110]. MIA-604 and MIA-690 (GH-RH antagonists) block the in vitro and in vivo growth of glioblastoma cells (U-87MG) [109]. Thus, both antagonists suppress tumor cell proliferation, reduce the size of tumor cells, downregulate cancer markers (e.g., mitogen-activated protein kinase 1 and Jun), upregulate tumor inhibitors (e.g., nexin and p53), block invasion and angiogenesis regulators (e.g., matrix metallopeptidase 1, angiopoietin 1, and S-100 calcium-binding protein), and favor autophagic and apoptotic mechanisms [109]. A previous study using different GH-RH antagonists (MIA-602 and JMR-132) showed similar results; both antagonists induced apoptosis in two glioblastoma cell lines (U-87MG and DBTRG 05) in in vitro and in vivo experiments [111]. These data demonstrate that GH-RH antagonists exert an antitumor action that affects multiple processes and significantly reduces the number and size of glioblastoma cells. However, compared to the administration of doxorubicin alone, the combination of the GH-RH agonist Jl-34 and the cytotoxic drug doxorubicin exerts a higher antitumor action against glioblastomas [112]. This agonist increases the inhibitory action mediated by doxorubicin on tumor cell proliferation, reduces the size of cells, favors apoptotic mechanisms, and decreases the release of humoral regulators (e.g., transforming growth factor-beta and fibroblast growth factor) of glial growth [112].

#### 2.3.2. Oxytocin

Oxytocin promotes the proliferation/viability of glioblastoma cells (U-87MG) [113]. The astrocytoma cell line MOG-G-UVW expresses oxytocin receptors, and oxytocin suppresses the proliferation of these cells, in which an increase in cAMP concentration has been observed [114]. Thus, oxytocin, via oxytocin receptors, regulates tumor growth by exerting a dual effect: proliferative or antiproliferative actions. Oxytocin receptor knockdown reduces the viability of glioblastoma U-87MG cells, and when these cells are treated with oxytocin, the expression of vimentin and drebrin, two cytoskeletal proteins, is increased [115]. This increase, mediated by oxytocin receptors, favors the formation/elongation of cell projections in U-87MG glioblastoma cells; these projections are counteracted in oxytocin receptor knockdown cells.

#### 2.3.3. Vasoactive Intestinal Peptide

The VIP system regulates the proliferation of glioblastoma cells [94]. VIP exerts an antiproliferative effect against serum-starved C6 glioma cells [116]; however, in another study, VIP promoted the proliferation of the latter cells, which was blocked by VIP antagonists [117]. VIP, probably through VPC2 receptors, blocks the proliferation of T98G glioblastoma cells [93]. However, the VIP antagonist VIPhyb blocks the proliferation of U-87MG, U-118MG, and U-373MG glioblastoma cell lines through PAC1 receptors in in vivo and in vitro experiments [118]. Thus, VIP exerts proliferative and antiproliferative effects on glioma cells.

Glioma cells expressing more VIP receptors are less invasive [90]. VIP agonists decrease the invasion of C6/U-87MG glioma cells, whereas VIP antagonists augment the migration and invasion of tumor cells [94]. This invasion is suppressed when PI3K/Akt and Sonic Hedgehog inhibitors are administered [94]. Moreover, VIP blocks the invasion of glioblastoma cells (U-87MG) exposed to hypoxia by regulating HIF and epidermal growth factor receptor (EGFR) expression, which are involved in cell migration/invasion and angiogenesis [119]. VPAC1 and VPAC2 receptors have been observed in the nuclei of glioblastoma cells [120]. VPAC1 receptors show strong nuclear staining, while VPAC2 receptors show weak staining. Importantly, a correlation between increased VPAC 1 receptor nuclear staining and glioma grade has been reported [120]. 

Table 1 and Figure 1 summarize the main findings regarding the involvement of oncogenic and anticancer peptides in glioma progression. It is important to note that tumor cells show distinctive characteristics such as energy reprogramming, immune destruction evasion, migration and invasion activation, cell death resistance, replicative immortality, and proliferative signal maintenance, as well as angiogenesis promotion in tumors. Thus, cancer cells escape from normal behavior, and importantly, peptidergic systems regulate the previously mentioned distinctive characteristics (Figure 2).

### 2.4. Other Peptides

#### 2.4.1. Galanin

The expression of galanin and galanin receptors (1 and 3) in human gliomas is known [122,123]. Galanin peptides and receptors have been reported in glioma-related macrophages and microglia; this suggests that the galaninergic system is present in the tumor microenvironment and that galanin can control the activity of these cells [122].

#### 2.4.2. Melanocyte-Stimulating Hormone

Alpha-melanocyte-stimulating hormone (MSH) suppresses the synthesis of pro-inflammatory cytokines and blocks the activation of NF-κB in glioma cells [124]. This means that the anti-inflammatory effects mediated by MSH are due to the control exerted on the activation of NF-κB. It seems that the anti-inflammatory effect exerted by MSH is mediated by the melanocortin 1 receptor [125].

#### 2.4.3. Neuropeptide Y

Neuropeptide Y (NPY) 1/2 receptors are expressed in glioblastoma cells [126,127]. Intratumoral nerve fibers containing NPY have been observed in glioblastomas, suggesting that this peptide, via NPY 2 receptors, modulates the activity of glioblastoma cells [126]. NPY increases the level of free intracellular Ca^++^ and decreases the cAMP (cyclic adenosine monophosphate) level in glioblastoma LN319 cells, which exclusively express the NPY 2 receptor [128]. NPY protein expression and NPY release from tumor cells significantly increase in C6 glioma cells after treatment with glutamate [129]. [Asn^6^, Pro^34^] NPY, a selective ligand of the NPY 1 receptor, ameliorates the therapeutic action of doxorubicin against gliomas, increasing the survival rate [126]. This ligand is delivered using nanomicelles; this strategy opens the door for the diagnosis and treatment of gliomas overexpressing the NPY 1 receptor.

## 3. Discussion

The overexpression of peptide receptors in tumor cells is crucial when specific peptide receptor ligands are used in nuclear medicine for research, diagnosis, management, and follow-up of gliomas and other tumors. This overexpression is of huge value for the treatment of tumors after applying cytotoxic peptide conjugate-based cancer therapy (e.g., peptide-drug conjugated) or targeted peptide receptor radionuclide therapy using radiopharmaceutical compounds (e.g., somatostatin analogs such as depreotide, ^68^Ga-DOTATATE (netspot), or [^177^Lu]Lu-DOTA-TATE (lutathera)) [87,130,131,132,133,134,135]. Importantly, a tumor-to-normal peptide receptor expression ratio of 3/1 or higher is needed to deliver a specific drug into cancer cells while healthy cells are spared; this significantly reduces adverse effects. Many peptidergic systems are involved in glioma development; hence, the use of peptide receptor antagonists or agonists could be a useful therapeutic tool to treat gliomas [130]. Moreover, many peptidergic systems are upregulated in gliomas, which means that this upregulation (peptide and/or peptide receptor) could be used as a biomarker for the disease. In general, overexpression of peptide receptors in glioma cells has beneficial effects by promoting the proliferation, migration, and invasion of glioma cells facilitated by oncogenic peptides. Additionally, these peptides, through their respective receptors, mediate anti-apoptotic mechanisms in glioma cells. Numerous studies have shown that blocking the signals mediated by oncogenic peptides induces the death of glioma cells via apoptosis. This suggests that glioma cells become dependent on the signals mediated by oncogenic peptides. This is crucial for developing anti-glioma therapeutic strategies. However, depending on the peptidergic system, overexpression of peptide receptors is not always beneficial for glioma cells, as glioma cells expressing more VIP receptors show a lower invasive capacity [90]. VIP/PACAP agonists reduce the invasion of glioma cells, whereas VIP antagonists increase the migration/invasion of tumor cells [94]. Thus, VIP receptor overexpression counteracts the migration and invasion of glioma cells, and this overexpression could support the exploration of anti-glioma strategies. This finding deserves to be investigated in depth.

### 3.1. Oncogenic/Anticancer Peptides Favoring/Counteracting Glioma Development

The following peptides, through their respective receptors, promote glioma progression (proliferation, migration, invasion, and angiogenesis): AMD, angiotensin II, GRP, bradykinin, CCK, endothelin, neuromedin B, neurotensin, and substance P (Figure 1 and Table 1) [38,49,52,53,60,61,99]. In addition, high AMD expression is related to the malignancy grade [7,17]. GRP receptor knockdown promotes senescence in glioma cells [30,36]. Bradykinin 1 and 2 receptor expression increases when glioma cells are treated with bradykinin, and this peptide upregulates PD-L1 expression in tumor cells and macrophages [40,43,44,45,46]. These data suggest that peptide receptor antagonists (alone or a cocktail of antagonists) of the above-mentioned oncogenic peptides could exert beneficial anti-glioma effects.

To the contrary, other peptides exert anti-glioma effects (antiproliferative, antimigration, anti-invasion, and anti-angiogenic): angiotensin (1-7) fragment, carnosine, CRF, methionine-enkephalin, Gn-RH (goserelin acetate), LH-RH (AN-152), Tat-Cx43266-283, Tat-NTS, PACAP, PACAP-27, PACAP-38, and somatostatin (Figure 1 and Table 1) [69,78,83,85,88,92,93,94,95,101,108,116]. This means that they could be used as anti-glioma drugs. Angiotensin (1-7) counteracts edema formation and blood–brain barrier damage by restoring the expression of tight junction proteins, and the antitumor effect mediated by this fragment increases when a blocker (SP600125) of the JNK pathway is administered [69]. Carnosine blocks anaerobic glycolytic metabolism in glioma cells and specifically destroys tumor cells in co-cultures with fibroblasts [71,72,73,74,76]. CRF treatment is more effective in terms of survival than treatment with dexamethasone or temozolomide [79]. Methionine-enkephalin increases the cytotoxic activity of microglia against glioma cells, increasing its antitumor efficacy and phagocytosis capacity [84], and somatostatin inhibits the release of angiogenic vascular endothelial growth factor from glioma cells [102].

Unlike previous peptides that exclusively exert antiproliferative action or a proliferative effect on glioma cells, GH-RH, oxytocin, and VIP promote both oncogenic and anticancer effects on these cells [94,109,110,112,113,114,116,119]. It is also important to highlight that the opioid morphine and the opioid peptide analog biphalin, respectively, exert proliferative and antiproliferative effects against human glioma T98G cells [82]. The dual effect exerted by the previous peptides could be related to the glioma cell type studied, to the receptor type/particular G protein involved in these effects, or to the dimerization, oligomerization, and heterodimerization of peptide receptors, which could regulate receptor trafficking, signaling pathways, and pharmacological responses. This is an important point that must be investigated in the future, as the therapeutic efficacy could depend on the peptide receptor type expressed by glioma cells and the signaling pathways involved.

### 3.2. Signaling Pathways Involved in Gliomas

Table 2 shows the main signaling pathways involved in glioma development.

The angiotensin II type 1 receptor/CXCR4/NF-κB signaling pathway is involved in glioma progression, and SOX9 is the downstream target of telmisartan [21,22]. Bradykinin activates the PI3K/Akt, AP1, ERK1/2, and c-Jun signaling pathways and favors the translocation of phosphorylated STAT3 and acetylated SP-1 into the nucleus [39,41,42]. Bradykinin regulates *aquaporin 4* gene expression and activates the MEK1-ERK1/2-NF-κB signaling pathway [43]. Endothelin B receptors mediate the proliferation of oligodendroglioma cells through an ERK mechanism [52]. The Tat-NTS peptide suppresses the NF-κB signaling pathway [78], and GRP receptors, via a PI3K-dependent mechanism, mediate the proliferation of glioma cells [35]. When interfering with neurotensin 1 receptor expression, an anti-invasion effect occurs through the Jun/miR-494/SOCS6 axis in glioma cells [54]. The neurotensin/neurotensin 1 receptor system activates the NF-κB and mitogen-activated protein kinase pathways, leading to the expression of Wnt proteins, and neurotensin 1 receptor expression is increased by Wnt3a, an activator of the Wnt pathway, while it is reduced by iCRT3, a blocker of this pathway [55]. Inhibition of the neurotensin 1 receptor promotes apoptosis via the let-7a-3p/Bcl-w axis in glioma cells [56]. Neurotensin controls, via activation of the interleukin-8/ERK/CXXC1/STAT3 pathway, the stem-like traits of glioblastoma stem cells [121]. Internalization of neurotensin by the neurotensin 2 receptor activates the phosphorylation of ERK1/2 in glioma cells, which is blocked by the neurotensin 2 receptor antagonist levocabastine [136]. VIP reduces the migration/invasion of glioma cells via protein kinase A-dependent inhibition of the Sonic Hedgehog/GLl1 and PI3K/Akt signaling pathways [94]. VIP blocks the invasion of glioma cells exposed to hypoxia by regulating the expression of HIF and EGFR, which are involved in cell migration/invasion and angiogenesis [119]. Somatostatin inhibits the migration of glioma cells without affecting Rac/PI3K activity, which is essential for motility regulation [101]. Finally, MIA-604 and MIA-690 (GH-RH antagonists) downregulate cancer markers (e.g., mitogen-activated protein kinase 1 and Jun) [109], and MSH exerts anti-inflammatory effects via the activation of NF-κB in glioma cells [124]. These data support the involvement of the above signaling pathways as potential therapeutic targets against gliomas, emphasizing the need to strengthen this research direction in future studies.

### 3.3. Antitumor Therapeutic Strategies against Gliomas

According to that reported in the above sections, different anticancer therapeutic strategies to suppress glioma progression have been reported: peptides (e.g., carnosine, [Asn^6^, Pro^34^] NPY, angiotensin (1-7) fragment, CRF, methionine-enkephalin, Gn-RH, LH-RH, Tat-NTS peptide, PACAP, and somatostatin), peptide receptor antagonists (EMA401, A3E, telmisartan, RC-3095, SSR240612, HOE-140, BKM-570, CCK antagonists, BQ788, MIA-602, MIA-604, MIA-690, JMR-132, SR48692, VIPhyb, aprepitant, and L-733,060), monoclonal antibodies against peptides, ingenol mebutate, interleukin-1β, miR-155, IκB kinase complex inhibitors, doxazosin, endothelin-converting enzyme 1 inhibitors, Wnt/β-catenin pathway inhibitors, miR-29b-1/miR-129-3p upregulation, tachykinin 1 receptor silencing, oxytocin or neurotensin 1 receptor knockdown, and silencing of the truncated splicing variant of the somatostatin 5 receptor (Figure 3).

Monoclonal antibodies against AMD inhibit tumor growth and enhance the antitumor effect exerted by temozolomide (in temozolomide-resistant glioma samples, overexpression of AMD has been reported); this inhibition has also been observed when the tetracyclic diterpenoid ester ingenol mebutate was administered by inhibiting the effects mediated by AMD [7,137]. AMD counteracts the apoptotic mechanisms promoted by hypoxia in glioblastoma cells, but interleukin-1β favors apoptosis in glioblastoma cells that synthesize AMD [18]. A decrease in glioma cell proliferation/invasion and in the expression of angiotensin II type 1 receptor occurs in glioma cells overexpressing miR-155; similar antitumor effects have been observed when administering IκB kinase complex inhibitors [21]. The repurposing of drugs (e.g., doxazosin, an antihypertensive drug) controlling the renin–angiotensin system has been suggested to treat gliomas [25,26]. This drug promotes apoptosis in glioma cells, and other antihypertensive compounds and angiotensin system inhibitors exert beneficial effects on the prognosis of patients with recurrent glioblastoma treated with bevacizumab [26,27]. Endothelin-converting enzyme 1 increases glioma cell migration and invasion; hence, it is a potential anti-glioma therapeutic target [81]. Oxytocin receptor knockdown reduces the viability of glioma cells [115], neurotensin 1 receptor knockdown suppresses the growth and invasion of these cells [54], and the silencing of the truncated splicing variant of the somatostatin 5 receptor sensitizes glioblastoma cells to the antitumor action exerted by the somatostatin analog pasireotide [103].

Biphalin, an opioid peptide analog, decreases the proliferation of glioma cells [82], methionine-enkephalin promotes apoptosis in glioma cells [83], and the ligand of the NPY 1 receptor, [Asn^6^, Pro^34^] NPY, increases the survival rate [125]. Carnosine inhibits the proliferation, migration, and invasion of glioma cells and specifically destroys tumor cells in co-cultures with fibroblasts [71,72,73,74]. The angiotensin (1-7) fragment and CRF exert antitumor actions against gliomas [69,107]. Gn-RH agonists reduce the proliferation of glioma cells [85]. LH-RH analogs and GH-RH agonists promote apoptosis of glioma cells [88,112]. Tat-NTS peptide blocks the proliferation, migration, and invasion of glioma cells [107], PACAP exerts an antiproliferative effect against glioma cells and reduces invasion [92,93,94,116], and somatostatin suppresses the migration of glioma cells. Thus, these peptides can be used as anti-glioma drugs.

Peptide receptor antagonists are also promising anti-glioma drugs (Table 3 and Figure 4). EMA401 (angiotensin II type 2 receptor antagonist) inhibits angiogenesis and the proliferation/invasion of glioma cells, and its A3E derivative reduces tumor volume, inhibits tumor cell proliferation, and augments apoptotic mechanisms in gliomas [19]. Antitumor effects (antiproliferative (apoptosis), antimigration, and anti-invasion) have also been reported using the angiotensin II receptor blocker telmisartan [22]. GRP receptor antagonists exert an antitumor effect against glioma cells [36], and co-administration of the GRP receptor antagonist RC-3095 and temozolomide significantly decreases glioma growth in in vitro and in vivo experiments [138]. SSR240612 (bradykinin 1 receptor antagonist) and HOE-140 (bradykinin 2 receptor antagonist) favor the death of glioma cells via apoptosis and necrosis [39,139], and treatment with BKM-570 (bradykinin antagonist) and temozolomide significantly increases the antitumor action of temozolomide against glioma cells [140]. CCK antagonists promote apoptosis in glioma cells by increasing caspase 3 activity [49], and the antagonist of the ETB receptor BQ788 inhibits the proliferation of oligodendroglioma cells and increases survival [52]. GH-RH antagonists (MIA-602, MIA-604, MIA-690, and JMR-132) block the growth and invasion of glioma cells as well as angiogenesis, upregulate tumor inhibitors, and promote autophagic and apoptotic mechanisms [109,111]. Glioma growth is blocked when the neurotensin/neurotensin 1 receptor system or the Wnt/β-catenin pathway is pharmacologically blocked [55]. Blockade of the neurotensin 1 receptor by antagonists (e.g., SR48692) promotes apoptosis in glioma cells and sensitizes these cells to antinomycin D or doxorubicin [56]. Thus, the neurotensin 1 receptor protects glioma cells from apoptosis. Neurotensin, through the neurotensin 1 receptor, favors the proliferation of glioma cells by blocking miR-29b-1/miR-129-3p [141]. This means that tumor cell proliferation can be suppressed by blocking the neurotensin 1 receptor or by upregulating the expression of miR-29b-1/miR-129-3p. VIP antagonists (e.g., VIPhyb) block the proliferation of glioma cells [117,118]. Substance P, via NK-1R, favors the proliferation, migration, and invasion of glioma cells and angiogenesis [60,61]. Glioma cells express NK-1R and NK-1R antagonists (e.g., aprepitant and L-733,060) and inhibit all of the previous effects mediated by substance P by promoting apoptotic mechanisms in glioma cells [60,61]. NK-1R expression is essential for the viability of glioma cells; apoptotic and necrotic mechanisms have been observed in glioma cells after silencing of the tachykinin 1 receptor [57,59]. Accordingly, the substance P/NK-1R system is a promising antitumor target for treating gliomas. This plethora of therapeutic strategies against gliomas must be channeled to transfer the current promising results to clinical practice in the near future.

### 3.4. Peptidergic Systems as Glioma Biomarkers

The overexpression of peptide receptors can be used as a potential glioma biomarker (NK-1R, angiotensin receptor, ETA receptor, ETB receptor, NPY receptor, VIP receptor, PACAP receptor, and somatostatin receptor), and as previously mentioned, this overexpression is also useful for potential and promising anti-glioma therapeutic strategies [19,57,86,89,126,127]. ETA receptors are overexpressed in gliomas, and the expression of ETA in tumor cells and microvessels gradually increases from grade 2 to 4 [50,51]. Patients with glioma overexpress ETB receptors, which is related to decreased survival [144]. High expression of neurotensin/neurotensin 1 receptors is related to poor prognosis in patients with gliomas [53]. Tumor cells expressing more VIP receptors are less invasive [90]. A correlation between increased VPAC 1 receptor nuclear staining and glioma grade has been suggested, and high PACAP receptor expression is correlated with the proliferative and malignant potential of astrocytoma cells [97,120]. The expression of the somatostatin 1 receptor is associated with overall survival in patients with gliomas, as well as the expression of the somatostatin 2 receptor, because its expression correlates with the WHO grade [98]. Expression of the somatostatin 5 receptor in glioma cells and of the somatostatin 3 receptor in tumor microvessels increases gradually from grade 2 to 4; both expressions are negatively associated with patient outcomes [51]. Overexpression of the truncated splicing variant of the somatostatin 5 receptor is related to malignancy in glioma cells [103]. Thus, overexpression of peptide receptors could be used as a biomarker for gliomas, which opens a promising door for the diagnosis and treatment of gliomas overexpressing peptide receptors.

High expression of neuromedin B is associated with higher survival of patients with glioma [145], and the survival rate of patients with gliomas is poorer when the expression of GH-RH is lacking [110]. Circulating progastrin has been suggested as a prognostic biomarker after surgery in patients with glioblastomas treated with radiotherapy/chemotherapy [146]. High endothelin-converting enzyme 1 expression has been associated with glioma development because this enzyme increases tumor cell migration and invasion [81]. Thus, endothelin-converting enzyme 1 may be a marker of poor prognosis. AMD has been suggested as a biomarker for targeted glycolytic therapy in patients with glioma because these patients can be classified according to their different glycolytic metabolism levels [137]. The angiotensinogen rs5050 GC genotype is related to poor prognosis in patients with glioma; this germline genetic variant has been proposed as a biomarker [147]. The low angiotensinogen promoter methylation observed in patients with recurrent gliomas is related to nonresponse to bevacizumab combination treatment with chemotherapy [148]. This information is useful for understanding which patients will not benefit from this combination therapy.

## 4. Future Research, Perspectives, and Conclusions

Glioma cells overexpress peptide receptors, and this characteristic is useful for research, diagnosis, management, and treatment of the disease. This overexpression serves to deliver chemotherapeutic drugs or compounds that induce apoptosis in glioma cells. It is important to know whether different peptidergic systems block glycolysis in glioma cells, as this process is crucial for cancer cells to obtain ATP. Many upregulated peptidergic systems are involved in glioma development. In addition, the above-mentioned overexpression of peptide receptors opens the door to developing anti-glioma research directions using peptide receptor agonists or antagonists. In this sense, the anticancer effect of other peptide receptor antagonists, rather than those that have already been tested,, must be studied and verified in preclinical studies, and the structure–function interactions between peptide receptors and peptides must be studied in detail to design new anti-glioma drugs. It is also important to remark on the important role that the tumor microenvironment plays in cancer progression. This must be investigated in depth in the future, as a full understanding of the tumor microenvironment will serve to develop new anti-glioma strategies and to understand how peptidergic systems regulate antitumor defenses. Peptides can activate/block immune cell activities and/or recruit these cells into the tumor, overcoming glioma cell immunosuppression [130].

Although many peptidergic systems have been studied in gliomas, the findings from other systems are fragmented, scarce, or absent, for example, regarding the involvement of peptide fragments in glioma progression. This is a basic research line that must be fully explored, with a focus on the expression of peptidergic systems that have not yet been studied in gliomas. Oncogenic peptidergic systems are potential and promising therapeutic targets for the use of peptide receptor antagonists to treat gliomas, whereas anticancer peptidergic systems can exert antitumor effects in gliomas through peptides or peptide receptor agonists. Peptide receptor antagonists have a higher therapeutic capacity than peptides/peptide receptor agonists, as peptides generally have a short half-life and poor bioavailability, although they usually show higher safety and solubility. However, several strategies to augment the therapeutic action of peptides, their delivery (oral administration), and stability have been developed, including peptide cyclization, cell-penetrating peptides, amino acid sequence manipulation, peptide-loaded nanoparticles, cell-targeting peptides, and peptide conjugation to polymers [11,87]. A recent review focused on the roles played by therapeutic peptide drugs against cancer; several peptide drugs have been approved by the FDA to treat some cancers (not gliomas), such as somatostatin or Gn-RH peptide analogs [87]. The synthesis and release of anticancer peptides could be an endogenous process that inhibits glioma progression; thus, the mechanisms regulating the synthesis and release of these peptides must be fully investigated and understood. In the same way, substances regulating the synthesis and release of peptides promoting glioma development must be investigated, as well as those compounds regulating the expression of peptide receptors. This knowledge is essential. The relationships and interactions between the different peptidergic systems involved in the progression/inhibition of gliomas must also be investigated in depth; it is important to understand the potential antitumor synergistic action mediated by two different peptide receptor antagonists promoting apoptosis in glioma cells, as well as the effects promoted by the combination of peptide receptor antagonists with peptide/peptide receptor agonists exerting anti-glioma actions. Basic research must be conducted to better understand the molecular mechanisms involved in glioma development and how signaling pathways interact when glioma cells are stimulated by several peptidergic systems exerting proliferative and antiproliferative effects. Thus, it is essential to know exactly which peptide receptors are involved in glioma progression/inhibition to develop drug–design studies and more specific ligands exerting antitumor effects against gliomas. Moreover, little information is available on the participation of endopeptidases in glioma progression, which is a promising research topic that must also be developed.

Many anticancer strategies for blocking glioma progression have been tested and confirmed. This plethora of anticancer strategies have shown beneficial effects against gliomas, including tumor growth inhibition, apoptosis, necrosis, decreased glioma cell migration/invasion, increased temozolomide/ionizing irradiation antitumor effect, decreased peptide receptor expression, angiogenesis blockade, increased survival, tumor inhibitor upregulation, increased autophagic mechanisms, and sensitization of glioma cells to the antitumor effects mediated by antinomycin D or doxorubicin. In addition, overexpression of peptide receptors serves as a tumor biomarker in gliomas; peptide receptor overexpression is related to malignant potential, decreased survival, and poor prognosis. However, overexpression of VIP receptors in glioma cells is related to a lower invasive capacity and high neuromedin B expression to higher survival of patients with glioma; however, the survival rate of these patients is poorer when the expression of GH-RH is lacking. Previous data have shown the functional complexity of the involvement of peptidergic systems in glioma progression and how it is regulated by many oncogenic and anticancer peptides. The angiotensinogen rs5050 GC genotype is related to poor prognosis in patients with glioma; however, much work must be conducted in this field and, in this sense, single-nucleotide polymorphisms regarding peptidergic systems must be fully investigated in gliomas. The epigenetic mechanisms regulating the peptidergic systems involved in glioma progression must be fully understood, as these mechanisms are related to the recurrence rate, carcinogenesis, gene expression, and peptide receptor expression. More experiments must be conducted using peptide receptor antagonists in combination therapy with radiotherapy or chemotherapy to understand whether an antitumor synergistic effect occurs against gliomas and whether the side effects of cytostatics are reduced.

In summation, many peptidergic systems (peptides/peptide receptors) are promising antitumor targets for treating gliomas using peptide receptor antagonists (EMA401, A3E, telmisartan, RC-3095, SSR240612, HOE-140, BKM-570, BQ788, MIA-602, MIA-604, MIA-690, JMR-132, SR48692, VIPhyb, aprepitant, and L-733,060) and/or peptide/peptide receptor agonists (carnosine, NPY, CRF, Gn-RH, LH-RH, PACAP, somatostatin, Tat-NTS peptide, angiotensin (1-7) fragment, and methionine-enkephalin). These data support the use of drug cocktails containing different compounds (e.g., peptide receptor antagonists/agonists) to treat gliomas. Moreover, the repurposing of drugs (antihypertensive and anti-emetic (aprepitant)) must be taken into consideration for the treatment of gliomas. Previous data support the development of clinical trials using, for example, peptide receptor antagonists alone or in combination with chemotherapy/radiotherapy. Peptidergic systems are diagnostic markers/therapeutic targets for the prognosis and treatment of patients with gliomas, increasing the possibilities for translational research; this line of research must be developed and enhanced in the future to finally establish clinical anti-glioma strategies. Unfortunately, no anticancer drug targeting peptidergic systems to treat gliomas is currently available in clinical practice despite the large amount of data supporting the involvement of peptides in glioma progression; however, therapeutic peptides have been approved by the FDA to treat other cancers. A promising finding is that carnosine specifically destroyed glioma cells in co-cultures with fibroblasts; meanwhile, in that study, the fibroblasts were healthy and survived [74].

## Figures and Tables

**Figure 1 ijms-25-07990-f001:**
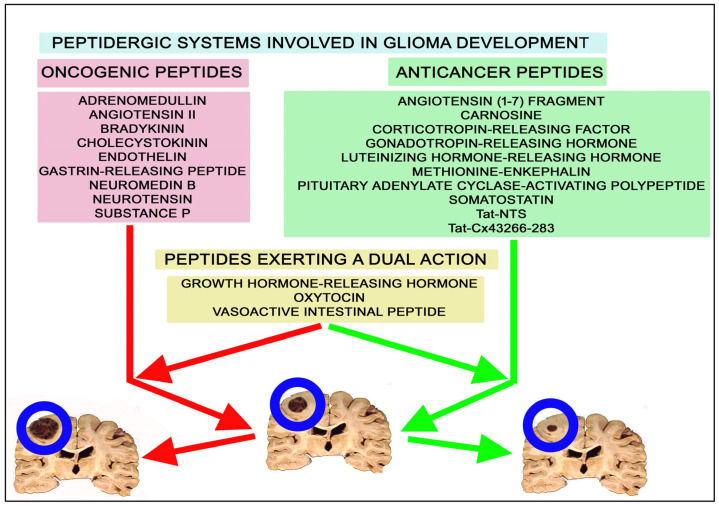
Oncogenic and anticancer peptides involved in glioma development.

**Figure 2 ijms-25-07990-f002:**
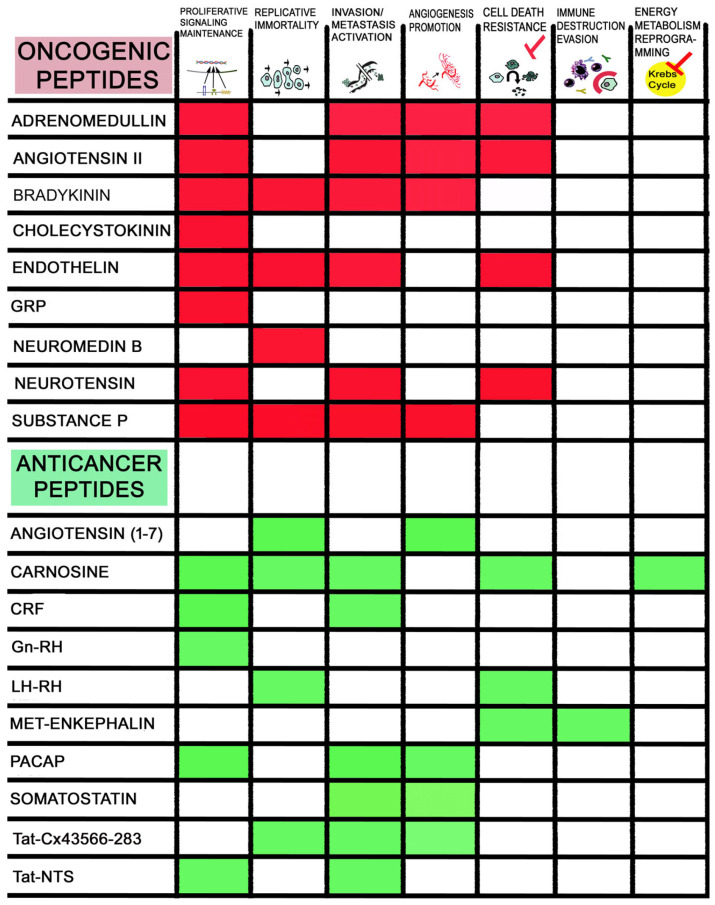
Oncogenic peptides favoring (red rectangles) and anticancer peptides counteracting (green rectangles) the hallmarks (proliferative signaling maintenance, replicative immortality, invasion and metastasis activation, angiogenesis promotion, cell death resistance, immune destruction evasion, and energy metabolism reprogramming) responsible for glioma development.

**Figure 3 ijms-25-07990-f003:**
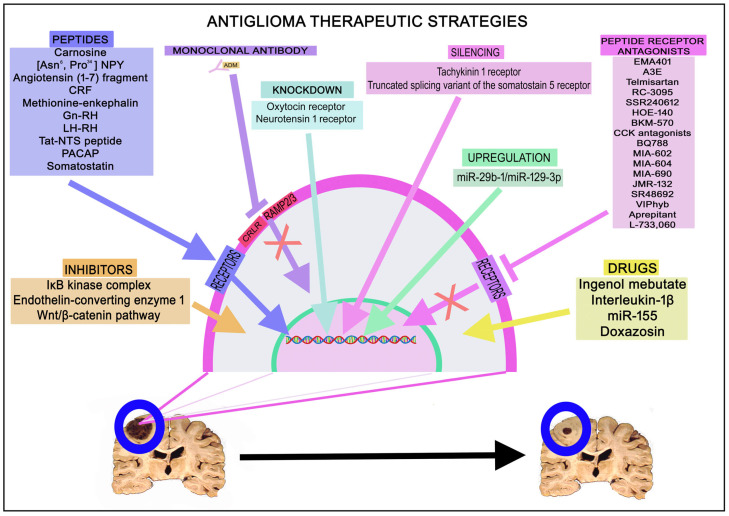
Anti-glioma therapeutic strategies: peptides, monoclonal antibodies, peptide receptor knockdown, peptide receptor antagonists, miR upregulation, drugs, and inhibitors.

**Figure 4 ijms-25-07990-f004:**
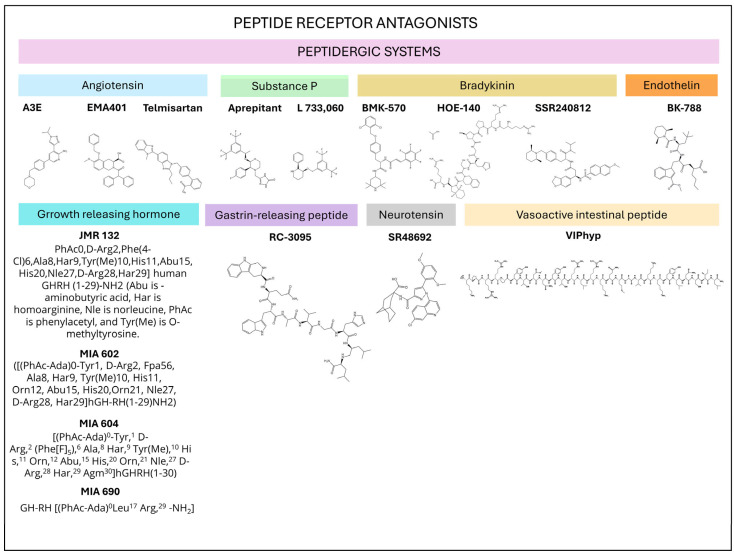
Chemical structures of peptide receptor antagonists showing anti-glioma effects. These structures were illustrated using KingDraw free software [142], except for VIPhyp, which was illustrated using the PepDraw program [143].

**Table 1 ijms-25-07990-t001:** Peptides favoring and/or counteracting glioma development.

Peptides Favoring Glioma Development		
Peptides	Actions	References
Adrenomedullin	Favors cell growth Its expression is positively associated with malignancy grade Increases cell invasive capacity/proliferation Favors angiogenesis Counteracts apoptosis	[7,16,17,18]
Angiotensin II	Cell proliferation/invasion associated with increased angiotensin II type 1 receptor expression	[21]
Bombesin/GRP	Favors cell proliferation and mitogen-activated protein pathway activation GRP receptors mediate cell proliferation GRP receptor knockdown induces cell senescence	[30,36]
Bradykinin	Favors cell proliferation Bradykinin 1 receptor mediates cell migration/invasion Bradykinin 1 receptor mediates glioblastoma and mesenchymal stem cell fusion	[39,40,41,42,43,48]
Cholecystokinin	Promotes cell growth through autocrine mechanisms	[49]
Endothelin	Endothelin B receptors mediate cell proliferation High endothelin-converting enzyme 1 expression associated with cell development Endothelin-converting enzyme 1: marker for poor prognosis	[52,81]
Neuromedin B	Increases *c-fos* gene expression Favors arachidonic acid release and cell growth	[38]
Neurotensin	Promotes cell proliferation/invasion Controls stem-like traits of glioblastoma stem cells High expression of neurotensin/neurotensin 1 receptor associated with poor prognosis	[53,54,121]
Substance P	Favors cell proliferation/migration/invasion and angiogenesis	[60,61]
Peptides Counteracting Glioma Development		
Peptides	Actions	References
Angiotensin (1-7)	Blocks cell growth and counteracts edema formation and blood–brain barrier damage	[69]
Carnosine	Blocks cell growth Inhibits cell proliferation/migration/invasion Prevents the formation of glioma cell colonies and specifically destroys tumor cells in co-cultures with fibroblasts Blocks cell anaerobic glycolytic metabolism	[71,72,73,74,76]
Corticotropin-Releasing Factor	Inhibits cell proliferation/invasion	[78]
Gonadotropin-Releasing Hormone	Gn-RH analogs decrease cell proliferation	[85]
Luteinizing Hormone-Releasing Hormone	LH-RH analogs promote apoptosis	[88]
Methionine-enkephalin	Promotes apoptosis Increases microglia cytotoxic activity/phagocytosis capacity against glioblastoma cells	[83,84]
Pituitary Adenylate Cyclase-Activating Polypeptide	High PACAP receptor expression correlated with cell proliferative/malignant potential Antiproliferative effect Blocks blood vessel formation and decreases vascular endothelial growth factor release	[92,95,97]
Somatostatin	Suppresses cell migration Inhibits vascular endothelial growth factor synthesis/release from glioma cells Somatostatin 5 receptor overexpression associated with increased cell proliferation/migration, recurrence, poor overall survival, and alteration in signaling pathways involved in tumor progression/aggressiveness	[101,102,103]
Tat-Cx43266-283	Antitumor effect	[108]
Tat-NTS	Inhibits cell proliferation/migration/invasion	[107]
Peptides Exerting A Dual Action		
Peptides	Actions	References
Growth Hormone-Releasing Hormone	Proliferative effect Antiproliferative action	[109,112]
Oxytocin	Promotes cell proliferation/viability Suppresses cell proliferation	[113,114]
Vasoactive Intestinal Peptide	Antiproliferative effect Promotes cell proliferation Blocks cell invasion	[116,117,119]

**Table 2 ijms-25-07990-t002:** Signaling pathways involved in glioma development.

Peptidergic System	Signaling Pathway	References
AMD	Akt, ERK1/2, Bax, Bcl-2 Calcitonin receptor-like receptor/receptor activity-modifying protein 2 and 3 STAT-3 activation	[15,16,17]
Angiotensin II	Angiotensin II type 1 receptor, CXCR4, NF-κB SOX9: telmisartan downstream target	[21,22]
Bradykinin	PI3K, Akt, AP1, c-Jun MEK1, ERK1/2, NF-κB Phosphorylated STAT3/acetylated SP-1 translocation into the nucleus *Aquaporin 4* gene expression	[39,41,42,43]
Endothelin	ERK	[52]
GH-RH	Mitogen-activated protein kinase 1, Jun	[109]
GRP	PI3K	[35]
Neurotensin	Jun, miR-494, SOCS6 NF-κB, mitogen-activated protein kinase Wnt Let-7a-3p/Bcl-w ERK1/2	[54,55,56,136]
Somatostatin	Rac, PI3K	[101]
Tat-NTS	NF-κB Matrix metalloproteinase downregulation	[107]
VIP	Protein kinase A, Sonic Hedgehog, GLl, PI3K, Akt HIF, EGFR	[94,119]

**Table 3 ijms-25-07990-t003:** Peptide receptor antagonists/agonists exerting anti-glioma action.

Peptide Receptor Antagonists	Targets	Peptidergic Systems	References
A3E	Angiotensin II type 2 receptor	Angiotensin	[19]
Aprepitant	Neurokinin-1 receptor	Substance P	[59,61]
BKM-570	Bradykinin receptor	Bradykinin	[140]
BQ788	Endothelin B receptor	Endothelin	[52]
EMA401 (Olodanrigan)	Angiotensin II type 2 receptor	Angiotensin	[19]
HOE-140 (Icatibant acetate; Firazyr)	Bradykinin 2 receptor	Bradykinin	[139]
JMR-132	Growth hormone-releasing hormone receptor	Growth hormone-releasing hormone	[111]
L-733,060	Neurokinin-1 receptor	Substance P	[60]
MIA-602	Growth hormone-releasing hormone receptor	Growth hormone-releasing hormone	[111]
MIA-604	Growth hormone-releasing hormone receptor	Growth hormone-releasing hormone	[109]
MIA-690	Growth hormone-releasing hormone receptor	Growth hormone-releasing hormone	[109]
RC-3095	Gastrin-releasing peptide receptor	Gastrin-releasing peptide	[138]
SR48692 (Meclinertant)	Neurotensin 1 receptor	Neurotensin	[56]
SSR240612	Bradykinin 1 receptor	Bradykinin	[139]
Telmisartan	Angiotensin II receptor	Angiotensin	[22]
VIPhyb	Vasoactive intestinal peptide receptor	Vasoactive intestinal peptide	[118]
Peptide Receptor Agonists	Targets	Peptidergic Systems	References
[Asn^6^, Pro^34^] NPY	Neuropeptide Y 1 receptor	Neuropeptide Y	[126]
Pasireotide	Somatostatin 5 receptor	Somatostatin	[103]

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
