# Peer review of "Glioma and Peptidergic Systems: Oncogenic and Anticancer Peptides"

_ijms, 2024, doi:10.3390/ijms25147990_

Round 1

Reviewer 1 Report

Comments and Suggestions for Authors

This review is focused on peptidergic systems in the settings of gliomas.

It is a great idea to propose a review on this topic as there is no one so far, to the best of my knowledge, whereas there are indeed a lot of published data, which could be useful to design new therapeutic strategies in the future. Nevertheless, the manuscript is much too longer, relatively bad-written and many paragraphs are confusing and redundant. For instance, influence of VIP on tumor growth is discussed lines 473-496, lines 672-677, lines 732-794, line 881, lines 907-908, lines 1030-1037. All these paragraphs contain redundant data. Rather than presenting a catalog of all peptides by alphabetic order, it would have been smarter to propose CONCISE paragraph about peptides increasing / decreasing / having a dual action on tumor growth. It is also the case for many other peptides. More relevant references have to be chosen, especially in the introduction.  Yet, if EXTENSIVE editing, clarification and REWRITING (not limited to the few points that have been highlighted below) is performed, this review should deserve publication.

 Minor remarks

11. The language has to be edited by a native speaker. For instance, in the first sentence of the abstract, there is a grammatical mistake (different peptide receptors = plural / is = singular), unless “tool” refers to the expression of different peptide receptors but if that is the case, the sentence is incorrect anyway as “tool” is not correctly used in this context).

22. The goal of the review is not mentioned, (neither in the abstract, nor in the introduction). The abstract is rather a catalog of peptides and mechanisms than a true abstract and needs to be rewritten.

33. Since 3 years (2021 WHO classification), glioma grade must not be expressed in roman numbers anymore but in Arabic numbers.

44. Introduction: the indicated overall survival in glioblastoma is not correct. Relevant data can be found in neuroepidemiological papers of reference (Ostrom et al).

55. Please, correct this sentence that does not make sense: “recurrence following treatments is very elevated; most of patients show recurrence within six months”.

66. Regarding glioblastoma management, TTF fields represent now a validated option and have to be cited.

77. The paragraph regarding the link between HIF1, IL-1B and AMD is very confusing. Conversely to what the authors state, they do not give explanation sustaining that IL1B promote apoptosis in GBM cell synthetizing AMD.

88.Line 229-230: redundancy of high/higher.

99. Line 266-267: “favoring the spread of glioma cells throughout the body”. I am not sure to understand. Metastases from glioma are really exceptional. Do the authors refer to the brain rather than the body?

110.  Line 271-274: which mesenchymal cells are the authors talking about? Mesenchymal signature is indeed one of the 4 signatures identified in glioblastoma cells. Interpretation of these data is lacking.

111.   Line 286-289: to what extent is there a link between the fact that carnosine augmented the efficacy of radiation and the fact that carnosine counteracted the side effects triggered by radiation?

112.   Line 303-304: the authors first explain that CRF receptor expression is increased in glioma and the sentence after that it is decreased. This part really needs clarification.

113.   It would have been interesting to discuss this paper: 10.1016/j.esmoop.2023.101626

114.   Figure 3 is not really reader-friendly as it contains a lot of abbreviation, particularly regarding receptors antagonists.

115.   Line 938-946: I do not understand to what extent this paragraph is related to the peptidergic systems.

116.   Line 949: “It seems that targeting these receptors the outcome of glioma patients could be improved.” This sentence, as many others, is grammatically incorrect.

117.   Line 978-987: all these data have already been presented earlier!!!!

Comments on the Quality of English Language

Very difficult to read

Author Response

Thank you very much for your e-mail concerning the review of the manuscript. We found the suggestions of the referees very helpful in improving the article. According to these suggestions the paper has been adjusted as follows (in the new Ms corrections appear in red):

Reviewer 1

This review is focused on peptidergic systems in the settings of gliomas. It is a great idea to propose a review on this topic as there is no one so far, to the best of my knowledge, whereas there are indeed a lot of published data, which could be useful to design new therapeutic strategies in the future. Nevertheless, the manuscript is much too longer, relatively bad-written and many paragraphs are confusing and redundant. For instance, influence of VIP on tumor growth is discussed lines 473-496, lines 672- 677, lines 732-794, line 881, lines 907-908, lines 1030-1037. All these paragraphs contain redundant data. Rather than presenting a catalog of all peptides by alphabetic order, it would have been smarter to propose CONCISE paragraph about peptides increasing / decreasing / having a dual action on tumor growth. It is also the case for many other peptides. More relevant references have to be chosen, especially in the introduction. Yet, if EXTENSIVE editing, clarification and REWRITING (not limited to the few points that have been highlighted below) is performed, this review should deserve publication.

The manuscript has been shortened (sections 2, 3 and 4), redundant data have been deleted and an extensive editing by the MDPI Author Services has been performed.

Section 2 has been restructured according to the suggestion of the reviewer: peptides increasing/decreasing/having a dual action on glioma. In this section, the effects of peptides on tumor cell proliferation/migration and angiogenesis and the expression of peptide receptors by glioma cells have been mentioned. See pages 2-14.

Section 3 has also been partially restructured. See lines 482-544.

New references have been added in Introduction: Ostrom et al. 2023 (reference 8), Szklener et al., 2023 (reference 9) and Vymazal et al., 2023 (reference 10) (see also below). See page 1, line 45 and page 2, line 48. Accordingly, references have been renumbered. See page 23 (list of references).

Minor remarks

  1. The language has to be edited by a native speaker. For instance, in the first sentence of the abstract, there is a grammatical mistake (different peptide receptors = plural / is = singular), unless “tool” refers to the expression of different peptide receptors but if that is the case, the sentence is incorrect anyway as “tool” is not correctly used in this context).

As indicated above, English have been checked by the MDPI Author Services.

  1. The goal of the review is not mentioned, (neither in the abstract, nor in the introduction). The abstract is rather a catalog of peptides and mechanisms than a true abstract and needs to be rewritten.

The aim of the review is mentioned. See lines 54-56.

The abstract has been rewritten. See page 1.

  1. Since 3 years (2021 WHO classification), glioma grade must not be expressed in roman numbers anymore but in Arabic numbers.

This has been corrected. See lines 42, 43, 194, 214, 296, 672 and 682.

  1. Introduction: the indicated overall survival in glioblastoma is not correct. Relevant data can be found in neuroepidemiological papers of reference (Ostrom et al).

As previously mentioned, this paper (reference 8) has been cited. See page 1, line 45.

Due to shortening the text, the sentence including the overall survival has been deleted in the new version. This also occurs in some points mentioned below.

  1. Please, correct this sentence that does not make sense: “recurrence following treatments is very elevated; most of patients show recurrence within six months”.

This sentence has been deleted in the new version.

  1. Regarding glioblastoma management, TTF fields represent now a validated option and have to be cited.

As previously mentioned, two references (Szklener et al., Front Oncol 2023, Vymazal et al., Front Oncol. 2023) have been cited (references 9 and 10). See page 2, line 48.

  1. The paragraph regarding the link between HIF1, IL-1B and AMD is very confusing. Conversely to what the authors state, they do not give explanation sustaining that IL1B promote apoptosis in GBM cell synthetizing AMD.

This has been better explained. See lines 88-95.

88.Line 229-230: redundancy of high/higher.

This sentence has been deleted in the new version.

  1. Line 266-267: “favoring the spread of glioma cells throughout the body”. I am not sure to understand. Metastases from glioma are really exceptional. Do the authors refer to the brain rather than the body?

This sentence has been corrected. See lines 170-172.

  1. Line 271-274: which mesenchymal cells are the authors talking about? Mesenchymal signature is indeed one of the 4 signatures identified in glioblastoma cells. Interpretation of these data is lacking.

New information has been added. See lines 175-182.

  1. Line 286-289: to what extent is there a link between the fact that carnosine augmented the efficacy of radiation and the fact that carnosine counteracted the side effects triggered by radiation?

Carnosine increases radiation efficacy on tumor cells and, at the same time, protects non-tumor cells from this radiation. Thus, carnosine reduces the side-effects mediated by radiation. See lines 271-274.

  1. Line 303-304: the authors first explain that CRF receptor expression is increased in glioma and the sentence after that it is decreased. This part really needs clarification.

Glioma express both the peptide CRF and CRF receptors. A downregulation was found in the mRNA level of the peptide (CRF mRNA level). See lines 278-281.

  1. It would have been interesting to discuss this paper: 10.1016/j.esmoop.2023.101626

This paper has been cited (reference 146). See lines 690-692 and page 32.

  1. Figure 3 is not really reader-friendly as it contains a lot of abbreviation, particularly regarding receptors antagonists.

These peptide receptors antagonists have been mentioned along the text and, in general, they do not have commercial names, except some of them. These commercial names appear in the new Table 3. See page 19.

  1. Line 938-946: I do not understand to what extent this paragraph is related to the peptidergic systems.

It has been deleted in the new version.

  1. Line 949: “It seems that targeting these receptors the outcome of glioma patients could be improved.” This sentence, as many others, is grammatically incorrect.

The sentence has been deleted.

  1. Line 978-987: all these data have already been presented earlier!!!!

It has been deleted.

Reviewer 2 Report

Comments and Suggestions for Authors

In this manuscript Sanchez M.L: et al, examined the involvement of both oncogenic and antitumor peptides in the onset and progression of glioma. Details of aberrant cellular pathways and the effects of specific drugs targeting oncogenic peptides are reported. Overall, this is a very interesting article, I would like to make some suggestions to improve its clarity:

-as regards the drugs mentioned, figures showing their chemical structure would be very useful. Furthermore, a final table showing the drugs mentioned, their targets (please be as detailed as possible here) and the modulated peptide would be helpful.

-the discussion is very long, and often repeats concepts reported in section 2. I highly recommend reducing this section quite a bit. Section 4 is also too long, should be shortened and could probably be more focused on a critical evaluation of those drugs capable of modulating the peptidergic system in glioma cells, with particular emphasis on in vivo evaluations. Table 3 can be deleted.

Author Response

Thank you very much for your e-mail concerning the review of the manuscript. We found the suggestions of the referees very helpful in improving the article. According to these suggestions the paper has been adjusted as follows (in the new Ms corrections appear in red):

Reviewer 2

In this manuscript Sanchez M.L: et al, examined the involvement of both oncogenic and antitumor peptides in the onset and progression of glioma. Details of aberrant cellular pathways and the effects of specific drugs targeting oncogenic peptides are reported. Overall, this is a very interesting article, I would like to make some suggestions to improve its clarity: -as regards the drugs mentioned, figures showing their chemical structure would be very useful. Furthermore, a final table showing the drugs mentioned, their targets (please be as detailed as possible here) and the modulated peptide would be helpful.

According to the suggestion of the reviewer, a new Figure (number 4) showing the chemical structures of drugs has been added as well as a new Table (number 3). See pages 19 and 20.

Section 3 has been partially restructured. See lines See lines 482-544.

-the discussion is very long, and often repeats concepts reported in section 2. I highly recommend reducing this section quite a bit. Section 4 is also too long, should be shortened and could probably be more focused on a critical evaluation of those drugs capable of modulating the peptidergic system in glioma cells, with particular emphasis on in vivo evaluations. Table 3 can be deleted.

The Manuscript has been shortened and repeated concepts have been avoided. Sections 2 and 4 have been shortened. Section 2 has been also restructured. See pages 2-14.

New Figure 4 and Table 3 are focused on drugs regulating glioma progression. See pages 19 and 20.

Old Table 3 has been deleted.

Round 2

Reviewer 1 Report

Comments and Suggestions for Authors

I recognize the efforts provided by the authors. Every remark has been seriously taken into account and, to me, the quality of the manuscript is far better.

I only have a minor remark: line 279-280 "CRF mRNA levels are downregulated in gliomas, and this factor inhibits the proliferation of glioma cells and favors long non-coding RNA-p21 expression, which suppresses both the proliferation and invasion of glioma cells".

The first part of the sentence is not very intuitive. I would propose to correct: CRF inhibits the proliferation of glioma cells and favors long non-coding RNA-p21 expression. In gliomas, CRF mRNA levels are downregulated, leading to a suppression of both the proliferation and invasion of glioma cells.

Author Response

Thank you very much for your e-mail concerning the review of the manuscript. According to the suggestion of Reviewer 1 the paper has been adjusted as follows (in the new Ms the correction appears in red):

I recognize the efforts provided by the authors. Every remark has been seriously taken into account and, to me, the quality of the manuscript is far better.

I only have a minor remark: line 279-280 "CRF mRNA levels are downregulated in gliomas, and this factor inhibits the proliferation of glioma cells and favors long non-coding RNA-p21 expression, which suppresses both the proliferation and invasion of glioma cells".

The first part of the sentence is not very intuitive. I would propose to correct:

CRF inhibits the proliferation of glioma cells and favors long non-coding RNA-p21 expression. In gliomas, CRF mRNA levels are downregulated, leading to a suppression of both the proliferation and invasion of glioma cells.

Thanks for your commentary. The above sentence has been changed. The new sentence appears in lines 280-283:

In gliomas, CRF mRNA levels are downregulated [78]. CRF inhibits the proliferation of glioma cells and favors long non-coding RNA-p21 expression; this expression leads to a suppression of both the proliferation and invasion of glioma cells [78].

We have slightly modified the sentence suggested by the reviewer, since we want to highlight that it is the long non-coding RNA that produces the inhibition of both proliferation and invasion.
